# Anti-Inflammatory and Neuroprotective Mechanisms of GTS-21, an α7 Nicotinic Acetylcholine Receptor Agonist, in Neuroinflammation and Parkinson’s Disease Mouse Models

**DOI:** 10.3390/ijms23084420

**Published:** 2022-04-16

**Authors:** Jung-Eun Park, Yea-Hyun Leem, Jin-Sun Park, Do-Yeon Kim, Jihee Lee Kang, Hee-Sun Kim

**Affiliations:** 1Department of Molecular Medicine, Inflammation-Cancer Microenvironment Research Center, School of Medicine, Ewha Womans University, Seoul 07804, Korea; jungeun17@ewhain.net (J.-E.P.); leemyy@ewha.ac.kr (Y.-H.L.); jsp@ewha.ac.kr (J.-S.P.); doyoun1531@naver.com (D.-Y.K.); 2Department of Physiology, Inflammation-Cancer Microenvironment Research Center, School of Medicine, Ewha Womans University, Seoul 07804, Korea; jihee@ewha.ac.kr

**Keywords:** α7 nAChR agonist, GTS-21, microglia, neuroinflammation, Parkinson’s disease, molecular mechanism

## Abstract

Neuroinflammation is crucial in the progression of neurodegenerative diseases. Thus, controlling neuroinflammation has been proposed as an important therapeutic strategy for neurodegenerative disease. In the present study, we examined the anti-inflammatory and neuroprotective effects of GTS-21, a selective α7 nicotinic acetylcholine receptor (α7 nAChR) agonist, in neuroinflammation and Parkinson’s disease (PD) mouse models. GTS-21 inhibited the expression of inducible nitric oxide synthase (iNOS) and proinflammatory cytokines in lipopolysaccharide (LPS)-stimulated BV2 microglial cells and primary microglia. Further research revealed that GTS-21 has anti-inflammatory properties by inhibiting PI3K/Akt, NF-κB, and upregulating AMPK, Nrf2, CREB, and PPARγ signals. The effects of GTS-21 on these pro-/anti-inflammatory signaling molecules were reversed by treatment with an α7 nAChR antagonist, suggesting that the anti-inflammatory effects of GTS-21 are mediated through α7 nAChR activation. The anti-inflammatory and neuroprotective properties of GTS-21 were then confirmed in LPS-induced systemic inflammation and MPTP-induced PD model mice. In LPS-injected mouse brains, GTS-21 reduced microglial activation and production of proinflammatory markers. Furthermore, in the brains of MPTP-injected mice, GTS-21 restored locomotor activity and dopaminergic neuronal cell death while inhibiting microglial activation and pro-inflammatory gene expression. These findings suggest that GTS-21 has therapeutic potential in neuroinflammatory and neurodegenerative diseases such as PD.

## 1. Introduction

Neuroinflammation is a key factor in the onset and progression of neurodegenerative disorders such as Parkinson’s disease (PD), Alzheimer’s disease (AD), and amyotrophic lateral sclerosis [1,2]. Microglia are the brain’s main immune cells and play a role in neuroinflammatory processes. Microglia rapidly respond to the pathological changes in the brain and regulate neuronal activity by producing neuroprotective or neurotoxic factors, depending on the microglial activation phenotypes [3,4]. However, persistent and unresolved microglial activation causes neuronal cell death, leading to various neurodegenerative diseases [5,6]. PD is the second most common neurodegenerative disease, with progressive impairment of motor functions such as resting tremors, bradykinesia, and muscle rigidity [7]. Selective loss of nigrostriatal dopaminergic neurons, aberrant α-synuclein aggregation known as Lewy bodies, and persistent neuroinflammation are the primary pathogenic characteristics of PD [8,9]. Damaged dopaminergic neurons release aggregated α-synuclein, which subsequently activates microglia to produce proinflammatory and neurotoxic factors. This vicious cycle of microglial activation and neuronal cell death aggravates PD progression [10,11]. Controlling microglial activation has thus been proposed as a key therapeutic method for PD and other neurodegenerative diseases.

Nicotinic acetylcholine receptors (nAChRs) are ligand-gated ion channels involved in a variety of biological functions including learning, memory, locomotion, and anxiety [11]. nAChRs are found in neurons as well as non-neuronal cells including microglia and astrocytes in the brain [12,13]. The α7 nicotinic acetylcholine receptor (α7 nAChR) is a subtype of nAChR that participates in the cholinergic anti-inflammatory pathway via vagus nerve stimulation [14]. Previous studies have reported that activation of α7 nAChR attenuates the LPS-induced pro-inflammatory cytokine production in macrophages and microglial cells [15,16,17]. In addition, activation of α7 nAChR by nicotine inhibited TNF-α and IL-1β expression in microglia and inhibited microglial proliferation in ischemia/reperfusion rats [18]. Nicotine also showed neuroprotective effects by inhibiting astrocyte activation in the MPTP mouse model [19]. Because α7 nAChR activation has an anti-inflammatory effect in microglia and astrocytes, α7 nAChR agonists have been proposed as potential therapeutic agents for neuroinflammatory disorders [20,21].

GTS-21 (3-(2,4-dimethoxybenzylidene)-anabaseine, DMXBA), a synthetic benzylidene derivative of naturally occurring anabaseine, is one of the most potent α7 nAChR agonists [22]. It quickly crosses the blood–brain barrier and enhances cognitive behavior, and hence is suggested as a drug candidate against AD [22,23,24]. Several studies have reported that GTS-21 is a promising therapeutic agent for neuroinflammatory conditions. It has been documented that GTS-21 suppresses LPS-induced TNF-α expression in rat primary microglia [25], and downregulates proinflammatory genes by inhibiting NF-κB and activating the Nrf2 signaling pathway in LPS-stimulated astrocytes [26]. In addition, GTS-21 promoted microglial Aβ phagocytosis and attenuated cognitive impairment in an AD mouse model [27]. However, the detailed mechanisms underlying GTS-21’s anti-inflammatory and neuroprotective effects have yet to be clearly elucidated. Furthermore, no studies on GTS-21’s effects on systemic inflammation or MPTP-induced PD mice models have been published.

As a result, we investigated the pharmacological activities and molecular mechanisms of GTS-21 in neuroinflammation and PD mouse models in this study. We discovered that GTS-21 exerts anti-inflammatory effects in microglia by modulating multiple signaling pathways, such as AMPK, PI3K/Akt, NF-κB, Nrf2, and CREB/PPARγ. The anti-inflammatory and neuroprotective properties of GTS-21 were then confirmed in LPS-induced systemic inflammation and MPTP-induced PD model mice. Detailed mechanistic studies have shown that Nrf2 and CREB signaling pathways are commonly involved in the anti-inflammatory and neuroprotective effects of GTS-21. The findings imply that GTS-21 stimulation of the α7 nAChR may have therapeutic promise for PD and other neurodegenerative diseases characterized by microglial activation.

## 2. Results

### 2.1. GTS-21 Inhibited the Expression of iNOS, COX-2 and Pro-Inflammatory Cytokines While Increasing Anti-Inflammatory TGF-β in LPS-Stimulated Microglial Cells

BV2 microglial cells and primary microglia were stimulated with LPS in the presence or absence of GTS-21, and the concentrations of NO and cytokines released into the culture media were determined. In LPS-stimulated BV2 and primary microglial cells, GTS-21 significantly reduced the production of pro-inflammatory molecules such as NO, TNF-α, IL-1β, and IL-6 while increasing the anti-inflammatory cytokine TGF-β (Figure 1A). The effects of GTS-21 on protein and mRNA expression of pro-/anti-inflammatory molecules in LPS-stimulated BV2 cells were then investigated using western blot and RT-PCR. As shown in Figure 1B,C, TNF-α, iNOS, COX-2, IL-1β, and IL-6 protein and mRNA levels were reduced by GTS-21, while TGF-β levels increased. These data support the anti-inflammatory role of GTS-21 in activated microglia.

### 2.2. GTS-21 Suppressed the Phosphorylation of Akt and NF-κB Activity, While It Increased AMPK Phosphorylation in LPS-Stimulated BV2 Microglial Cells

To determine the anti-inflammatory mechanism of GTS-21, we first looked at its effect on MAP kinase phosphorylation. As shown in Figure 2A, the phosphorylation of the three types of MAPKs induced by LPS was unaffected by GTS-21. Next, we looked at how GTS-21 affected Akt and AMPK, which are important pro- and anti-inflammatory signaling molecules, respectively. In LPS-stimulated BV2 cells, GTS-21 inhibited Akt phosphorylation while increasing AMPK phosphorylation (Figure 2B,C). Furthermore, GTS-21 inhibited the DNA binding and transcriptional activities of NF-κB, a transcription factor that regulates the expression of cytokines and *iNOS* genes (Figure 2D,E). These findings suggest that GTS-21 inhibits inflammation by activating AMPK and inactivating the PI3K/Akt and NF-κB signaling pathways.

### 2.3. GTS-21 Reduced ROS Production by Modulating the NADPH Oxidase Subunit p47phox, While Increasing Antioxidant Enzyme Gene Expression via the Nrf2/ARE Signaling Pathway

Reactive oxygen species (ROS) are secondary messengers that activate pro-inflammatory pathways in microglia [28]. In BV2 cells and primary microglia, GTS-21 significantly inhibited LPS-induced ROS production (Figure 3A,B). Additionally, GTS-21 reduced the production of 4-hydroxy-2E-nonenal (HNE), an oxidative stress marker (Figure 3B). We evaluated the effect of GTS-21 on NADPH oxidase components since NADPH oxidase is a significant enzyme involved in ROS production. GTS-21 suppressed the phosphorylation and mRNA expression of the p47phox subunit without affecting other components (Figure 3C,D). The effects of GTS-21 on Nrf2/ARE signaling and its downstream antioxidant enzymes were then investigated. It increased Nrf2 DNA binding, nuclear translocation, and transcriptional activity in LPS-stimulated BV2 cells (Figure 3E–G). In accordance with this, GTS-21 increased the expression of antioxidant enzymes controlled by Nrf2/ARE signaling, such as HO-1, NQO1, and catalase (Figure 3H,I). These findings imply that the Nrf2/ARE signaling pathway is involved in the anti-inflammatory and antioxidant mechanisms of GTS-21.

### 2.4. GTS-21 Upregulated the CREB and PPAR-γ Signaling in LPS-Stimulated BV2 Cells

We then looked at how GTS-21 affected the transcription factors CREB and PPAR-γ, which mediate anti-inflammatory and antioxidant effects by association with Nrf2 [29,30]. As shown in Figure 4A–D, GTS-21 upregulated CREB signaling by increasing the phosphorylation, nuclear translocation, DNA binding, and reporter gene activities of CREB. In addition, we found that GTS-21 restored *PPAR-γ* expression at both the mRNA and protein levels., which was reduced by LPS treatment (Figure 4E,F). These findings suggest that the PKA/CREB and PPAR-γ signaling pathways are also involved in the anti-inflammatory mechanism of GTS-21 in LPS-stimulated BV2 microglial cells.

### 2.5. The α7 nAChR Antagonists, Methyllycaconitine and α-Bungarotoxin, Reversed the Effects of GTS-21 on Pro-/Anti-Inflammatory Signaling Molecules

To see if α7 nAChR mediates GTS-21’s anti-inflammatory effects, BV2 cells and primary microglia were treated with an α7 nAChR antagonist, methyllycaconitine (MLA) or α-bungarotoxin (α-BTX), before adding LPS and/or GTS-21. Treatment with MLA or α-BTX blocked GTS-21-mediated suppression of NO, TNF-α, IL-6, and ROS in LPS-stimulated BV2 cells and primary microglia, as shown in Figure 5A. In addition, MLA reversed the effect of GTS-21 on reporter gene activities of NF-κB, Nrf2, CREB, and PPAR-γ in LPS-stimulated BV2 cells (Figure 5B). Furthermore, MLA reversed GTS-21-mediated inhibition of p-Akt, and upregulation of p-AMPK and p-CREB (Figure 5C). These findings suggest that α7 nAChR mediates GTS-21’s anti-inflammatory effects in microglia by modulating pro- and anti-inflammatory signaling molecules.

### 2.6. GTS-21 Reduced Microglial Activation and Pro-Inflammatory Gene Expression in LPS-Injected Mouse Brains

To confirm the anti-inflammatory effects of GTS-21 in vivo, we examined its effects in the brains of LPS-injected mice. Prior to LPS administration, mice were injected with GTS-21. Microglial activation in the brains of LPS-injected mice was assessed 24 h after LPS injection. The number of Iba-1-positive cells in the prefrontal cortex, striatum, hippocampus, and substantia nigra increased in response to systemic LPS. GTS-21 reduced the number of these cells, implying that it inhibited microglial activation (Figure 6A,B; CTX, F_3, 12_ = 21.83, *p* < 0.01; ST, F_3, 12_ = 38.18, *p* < 0.01; DG, F_3, 12_ = 24.76, *p* < 0.01; SN, F_3, 12_ = 11.28, *p* < 0.01). In addition, GTS-21 reduced the expression of proinflammatory markers such as *TNF-α*, *iNOS*, *COX-2*, *IL-1β*, and *IL-6* in the cortex of LPS-injected mice while increasing the anti-inflammatory cytokine *TGF-β* (Figure 6C,D; *TNF-α*, F_3, 8_ = 129.81, *p* < 0.01; *iNOS*, F_3, 8_ = 44.82, *p* < 0.01; *COX-2*, F_3, 8_ = 71.35, *p* < 0.01; *IL-1β*, F_3, 8_ = 39.82, *p* < 0.01; *IL-6*, F_3, 8_ = 39.90, *p* < 0.01; *TGF-β*, F_3, 8_ = 9.96, P < 0.01). Moreover, GTS-21 upregulated the expression of Nrf2, HO-1, NQO1, and p-CREB (Figure 6E,F; Nrf2, F_3, 16_ = 5.68, *p* < 0.01; HO-1, F_3, 14_ = 3.96, *p* < 0.05; NQO1, F_3, 12_ = 7.79, *p* < 0.01; *p*-CREB, F_3, 12_ = 14.24, *p* < 0.01).

### 2.7. GTS-21 Exerted Neuroprotective and Anti-Inflammatory Effects in the MPTP-Induced PD Mouse Model

We investigated whether GTS-21 has neuroprotective and anti-inflammatory effects in an MPTP-induced PD mouse model. Mice were given GTS-21 intraperitoneally once a day for three days prior to MPTP injection and sacrificed seven days later for analysis. The experimental procedure is depicted schematically in Figure 7A. GTS-21 administration improved MPTP-induced decline of retention time on rods in the rotarod test, and shortened MPTP-induced extension of descending time in the pole test (Figure 7B,C; rotarod, F_3, 31_ = 14.62, *p* < 0.01; pole, F_3, 32_ = 13.69, *p* < 0.01). The data from rotarod and pole tests imply that GTS-21 can improve impaired locomotor activity. Immunohistochemistry using the tyrosine hydroxylase (TH) antibody showed that GTS-21 inhibited dopaminergic neuronal cell death in the substantia nigra and recovered dopaminergic neuronal fibers in the striatum (Figure 7D,E; SN, F_3, 40_ = 20.35, *p* < 0.01; striatum, F_3, 41_ = 26.20, *p* < 0.01). Western blot analysis supported these findings by demonstrating that MPTP injection reduced the level of TH, which was recovered by GTS-21. Moreover, MPTP injection decreased neuronal survival factors in the substantia nigra such as p-CREB, PGC-1α, BDNF, and Bcl2, which were all reversed by GTS-21 treatment (Figure 7F,G; TH, F_3, 13_ = 5.30, *p* < 0.05; p-CREB, F_3, 12_ = 4.59, *p* < 0.05; PGC-1α, F_3, 12_ = 9.85, *p* < 0.01; BDNF, F_3, 12_ = 11.01, *p* < 0.01; Bcl2, F_3, 12_ = 3.40, *p* < 0.05). The effect of GTS-21 on microglial activation in the brains of MPTP-injected mice was then investigated. As shown in Figure 8A–D, GTS-21 inhibited microglial activation in the substantia nigra and striatum (SN, F_3, 36_ = 22.97, *p* < 0.01; striatum F_3, 40_ = 27.37, *p* < 0.01) and also suppressed the expression of proinflammatory markers, such as TNF-α, iNOS, and IL-1β, induced by MPTP treatment (TNF-α, F_3, 12_ = 28.47, *p* < 0.01; IL-1β, F_3, 12_ = 8.38, *p* < 0.01; iNOS, F_3, 12_ = 8.30, *p* < 0.01). Moreover, GTS-21 recovered the protein levels of HO-1, NQO1, and its upstream Nrf2, which were reduced by MPTP treatment (Figure 8E,F; HO-1, F_3, 12_ = 9.84, *p* < 0.01; NQO1, F_3, 12_ = 2.82, *p* < 0.05; Nrf2, F_3, 12_ = 3.39, *p* < 0.05).

## 3. Discussion

The current study found that GTS-21 has anti-inflammatory and neuroprotective properties in LPS-induced neuroinflammation and MPTP-induced PD model mice. In LPS-stimulated microglia, GTS-21 reduced the expression of iNOS and proinflammatory cytokines while increasing anti-inflammatory TGF-β. Further molecular research demonstrated that GTS-21 exhibits anti-inflammatory effects by suppressing PI3K/Akt, NF-κB, and upregulating AMPK, Nrf2, CREB, and PPARγ signals. We confirmed the anti-inflammatory effect of GTS-21 under neuroinflammatory conditions in vivo. GTS-21 inhibited microglial activation and the production of pro-inflammatory markers and upregulated the expression of the antioxidant enzyme HO-1 and NQO1 in the brains of LPS-injected mice. In addition, we found that GTS-21 has neuroprotective and anti-inflammatory properties in PD mice. GTS-21 restored locomotor activity, reduced dopaminergic neuronal cell death, and attenuated microglial activation and proinflammatory marker expression in MPTP-injected mouse brains. These findings support the therapeutic potential of GTS-21 in neuroinflammatory disorders such as PD.

The α7 nAChRs play a role in synaptic plasticity, neuronal survival, and neuroprotection. Toward this, studies been reported that α7 nAChR agonists have the potential to treat neurodegenerative diseases, brain ischemia, schizophrenia, and pain [31]. In particular, α7 nAChRs are present on dopaminergic neuronal soma, and the protective effects of α7 nAChR agonists against dopaminergic neuronal cell death have been demonstrated in several PD animal models [11,19]. However, to date, the molecular mechanisms have not been fully elucidated. Moreover, the effect of GTS-21 in the MPTP-induced PD mouse model has not been reported. In the current study, we showed that GTS-21 has neuroprotective effects in MPTP-induced PD mice. GTS-21 restored MPTP-induced motor impairment, as shown by rotarod and pole tests, and recovered the loss of TH-positive dopaminergic neurons in the SN. In addition, GTS-21 upregulated the expression of p-CREB, PGC-1α, BDNF, and Bcl2, which are under the control of the PKA/CREB signaling pathway [29,32]. This study also found that GTS-21 upregulated the expression of Nrf2, which controls Bcl2 and BDNF expression and mediates anti-inflammatory activities in the brain [33,34]. Consistent with this, GTS-21 inhibited microglial activation and pro-inflammatory marker expression and upregulated HO-1/NQO1 expression with its upstream modulator Nrf2 in MPTP mice. These findings suggest that the PKA/CREB and Nrf2 signaling pathways are largely involved in the neuroprotective and anti-inflammatory mechanisms of GTS-21 in MPTP-induced PD mice.

Previous research has found that α7 nAChR regulates the innate immune system and has a function in the cholinergic anti-inflammatory pathway [20,21]. The anti-inflammatory role of α7 nAChR has been reported in several cell types such as human monocytes, macrophages, microglia, astrocytes, and disease animal models or patients with sepsis, rheumatoid arthritis, pancreatitis, and neuronal disorders including brain ischemia, AD, and PD [17,31,35,36,37,38,39]. The suppression of NF-κB and upregulation of the Nrf2/HO-1 signaling pathway have been suggested as the main mechanisms for the anti-inflammatory effect of α7 nAChR activation in vitro and in vivo [31]. To further analyze the molecular mechanism underlying the anti-inflammatory effects of α7 nAChR, we looked at how GTS-21 affected various signaling molecules in activated microglia. GTS-21 was found to have anti-inflammatory effects by inhibiting Akt phosphorylation and NF-κB activity and upregulating anti-inflammatory signals such as AMPK, Nrf2, CREB, and PPARγ. Our group recently demonstrated that AMPK acts as an upstream modulator leading to the inhibition of pro-inflammatory signals such as PI3K/Akt and NF-κB and enhancement of anti-inflammatory signals such as PKA/CREB and Nrf2/ARE in activated microglia [40,41]. Moreover, we demonstrated the crosstalk between CREB, Nrf2, and PPARγ signals in microglia. We showed that the PKA/CREB signaling pathway regulates transcription factors Nrf2 and PPARγ by acting as an upstream modulator [29], and PPARγ also controls Nrf2, CREB, and NF-κB signaling in activated microglia [30]. Taken together, the anti-inflammatory signals such as AMPK, CREB, Nrf2, and PPARγ appear to be interconnected, leading to strong inhibition of pro-inflammatory molecules in GTS-21-treated activated microglia.

In this study, we found that the α7 nAChR antagonists, MLA, and α-BTX, reversed the effects of GTS-21 on NO, ROS, and pro-inflammatory cytokines in LPS-stimulated BV2 cells and primary microglia. Moreover, treatment with MLA abolished the effects of GTS-21 on pro-inflammatory signaling molecules such as PI3K/Akt, NF-κB, and anti-inflammatory signaling molecules such as AMPK and Nrf2/CREB/PPARγ. The findings suggest that the anti-inflammatory effects of GTS-21 are at least partially mediated by α7 nAChR activation in microglia.

In summary, the present study demonstrated the anti-inflammatory, antioxidant, and neuroprotective effects of the α7 nAChR agonist GTS-21 in neuroinflammation and PD mouse models. Detailed mechanistic studies have revealed that multiple signaling pathways are involved in the anti-inflammatory and neuroprotective effects of GTS-21. Among them, the Nrf2/ARE and PKA/CREB pathways can be considered to be the common mechanisms that mediate both the anti-inflammatory and neuroprotective effects of GTS-21. Therefore, our data collectively support that activation of α7 nAChR by GTS-21 has therapeutic potential for PD and other neurodegenerative diseases that are accompanied by neuroinflammation.

## 4. Materials and Methods

### 4.1. Reagents and Antibodies

GTS-21, α-bungarotoxin, and antibodies against Bcl2 and PGC-1α were purchased from Abcam (Cambridge, UK). LPS (Escherichia coli serotype 055:B5), methyllycaconitine, and antibodies against β-actin and BDNF were purchased from Sigma-Aldrich (St. Louis, MO, USA). MPTP was obtained from Tokyo Chemical Industry Co. (Tokyo, Japan). Antibodies against Iba-1 were purchased from Wako (Osaka, Japan). Antibodies against phospho-/total forms of AMPK, Akt, MAP kinases, CREB, and antibodies for IL-6, TGF-β, NQO1, PPAR-γ, and TH were obtained from Cell Signaling Technology (Beverley, CA, USA). While antibodies against phospho-p47phox were provided by Assaybiotech (Sunnyvale, CA, USA), those against TNF-α, Nrf2, HO-1, catalase, lamin A, and COX-2 were obtained from Santa Cruz Biotechnology (Santa Cruz, CA, USA). Antibodies against iNOS and IL-1β were purchased from BD Biosciences (San Jose, CA, USA), and an antibody for 4-hydroxy-2E-nonenal (HNE) was purchased from Alpha Diagnostic International (San Antonio, TX, USA).

### 4.2. Culture of Microglial Cells

The immortalized murine BV2 microglial cell line [42] was grown in Dulbecco’s modified Eagle’s medium (DMEM) supplemented with 10% heat-inactivated FBS, streptomycin (10 μg/mL), and penicillin (10 U/mL) in an incubator with 5% CO_2_ at 37 °C. As previously described, primary microglial cells were cultured from the cerebral cortices of 1- to 2-day-old Sprague-Dawley rat pups [43]. The microglial cultures were more than 95% pure, as determined by western blot and immunocytochemistry with Iba-1 antibody (data not shown).

### 4.3. Measurement of Cytokines, Nitrite, and Intracellular ROS Levels

BV2 cells or primary microglia (1 × 10^5^ cells per well in a 24-well plate) were pre-treated with GTS-21 (1–50 μM) for 1 h before being stimulated with LPS (100 ng/mL for BV2, 10 ng/mL for primary microglia) and incubated for 16 h. The concentrations of TNF-α, IL-1β, IL-6, and TGF-β in culture media were determined using enzyme-linked immunosorbent assay (ELISA) kits (R&D Systems, Minneapolis, MN, USA). The Griess reagent was used to measure the accumulated nitrite oxide levels (Promega, Madison, WI, USA). H_2_DCF-DA (Sigma-Aldrich) and CellROX^TM^ green reagent (Thermo Fisher Scientific, Waltham, MA, USA) were used to detect intracellular ROS levels, as described before [44,45].

### 4.4. Mice

Adult male C57BL/6 mice (20–25 g; 8 weeks old) were purchased from Orient Bio Inc., a Charles River Laboratories branch in Seongnam, Korea. The mice were kept at 21°C on a 12-h light/dark cycle, with free access to water and rodent chow. Every effort was made to keep the animals as stress-free as possible. All experiments were carried out in accordance with the National Institutes of Health and Ewha Womans University guidelines for the care and use of laboratory animals. The Institutional Animal Care and Use Committee of the Medical School of Ewha Womans University approved the study (#EUM 20-022).

### 4.5. Drug Administration

Mice were randomly assigned to one of four groups (control, LPS, LPS + GTS-21, and GTS-21; N = 9 per group). GTS-21 was dissolved in saline and given daily (5 mg/kg, i.p.) for four days. As previously described, LPS (5 mg/kg, i.p.) was injected 1 h after the final GTS-21 administration [43]. C57BL/6 mice were divided into four groups to study the MPTP mouse model (control, MPTP, MPTP + GTS-21, and GTS-21; each group, N = 9). GTS-21 (2 mg/kg, i.p.) was given daily for three days in a row. On the day following the final GTS-21 administration, MPTP (20 mg/kg, i. p.) was injected four times at two-hour intervals [29].

### 4.6. Behavioral Test

Mice were given behavioral tests 2 and 6 days after receiving MPTP injections (for the rotarod test, 2 days; pole test, 6 days). An accelerated rotarod test was used to evaluate the mice’s motor coordination. Prior to the MPTP treatment, the mice were trained for four consecutive days, with three trials per day, each day with an intersession interval of 15 min. On test day, mice were placed on the resting drum (3-cm diameter) of a rotarod apparatus (Harvard Apparatus, Holliston, MA, USA) for at least 1 min. Over a 300-s period, the rotarod’s speed was increased from 4 to 40 rpm. The mice were subjected to three trials with 15 min intervals between trials. The retention time of the rod in each trial was recorded. A pole test (50 cm height, 0.5 cm diameter, 120 s) was used to assess akinesia. Initially, all mice were trained to successfully descend from the pole’s top to bottom. The time it took each mouse to descend the pole was recorded. Each mouse was subjected to three trials, and the average was recorded.

### 4.7. Preparation of Brain Tissue

The mice were anesthetized with sodium pentobarbital (80 mg/kg body weight, i.p.) before being perfused transcardially with 0.9% saline followed by 4% paraformaldehyde for tissue fixation. The brains were then isolated and cryoprotected in a 30% sucrose solution at 4 °C. Mice were transcardially perfused with saline for biochemical analysis. The striatum and substantia nigra were dissected from each brain according to the Paxinos mouse brain atlas and immediately frozen in liquid nitrogen until use.

### 4.8. Immunohistochemistry and Immunofluorescence Analysis

40-μm-thick coronal sections were cut with a cryotome (CM1860; Leica, Mannheim, Germany) and stored at −20 °C in an anti-freezing solution (30% ethylene glycol and 30% glycerol in phosphate-buffered saline). Sections were treated with 0.3% H_2_O_2_ for 30 min and 4% BSA for 1 h before IHC staining to inhibit endogenous peroxidation and prevent non-specific binding, respectively. Sections were incubated with primary antibodies overnight, then with biotinylated secondary antibodies for 1 h at 25 °C before being incubated with avidin–biotin–HRP complex reagent solution (Vector Laboratories, Burlingame, CA, USA). Following that, a peroxidase reaction was carried out using diaminobenzidine tetrahydrochloride (Vector Laboratories). BV2 cells were fixed with 4% paraformaldehyde for 15 min before being incubated with 2% BSA/3% horse serum for 30 min to block nonspecific antibody binding. The cells were then incubated with anti-HNE primary antibodies overnight, followed by fluorochrome-conjugated secondary antibodies (Alexa Fluor 488) for 1 h at 25 °C. A Leica DM750 microscope was used to capture digital images of the IHC and IF staining, and ImageJ software, version 1.37, was used for quantification (National Institutes of Health, Bethesda, MD, USA).

### 4.9. Reverse-Transcription Polymerase Chain Reaction (RT-PCR)

Total RNA from BV2 cells and mouse brain tissue was isolated using TRIzol reagent (Invitrogen, Carlsbad, CA, USA). 1 μg of RNA was reverse transcribed in a reaction mixture containing 1 U RNase inhibitor, 500 ng random primer, 3 mM MgCl_2_, 0.5 mM dNTP, 1X RT buffer, and 10 U reverse transcriptase (Promega, Madison, WI, USA) for RT-PCR. The cDNA was used as a template for the PCR reaction, which was carried out with Go Taq polymerase (Promega) and primers. The RT-PCR was performed on a Bio-Rad T100 thermal cycler (Bio-Rad, Richmond, CA, USA). The ABI PRISM 7000 Sequence Detection System (Applied Biosystems, Foster City, CA, USA) was used to perform quantitative RT-PCR using Sensi FAST^TM^ SYBR Hi-ROX Mix (Bioline, London, UK). The following formula was used to normalize the expression levels of the target genes towards GADPH: 2 ^(Ct, test gene-Ct, GAPDH)^. Table 1 lists the primer sequences utilized in the PCR experiments.

### 4.10. Western Blot Analysis

Whole-cell protein lysates and brain tissue homogenates were prepared in a lysis buffer containing a protease inhibitor cocktail (10 mM Tris [pH 7.4], 30 mM NaCl, 1% Triton X-100, 0.1% SDS, 0.1% sodium deoxycholate, and 1 mM EDTA). SDS-PAGE was used to separate protein samples, which were then transferred to a nitrocellulose membrane and incubated with primary antibodies overnight at 4 °C. After washing thoroughly with TBST, HRP-conjugated secondary antibodies (BioRad, Hercules, CA, USA 1:1000 dilution in TBST) were applied, and the blots were developed using an enhanced chemiluminescence detection kit (Thermo Fisher Scientific, Waltham, MA, USA). Using ImageJ software, the density of specific target bands was normalized against β-actin for quantification. The normalized band densities of triplicate or quadruple samples were averaged.

### 4.11. Transient Transfection and Luciferase Assay

BV2 cells were transfected with 1 μg of reporter plasmid DNA ([κB]_3_-luc, ARE-luc, CRE-luc, or PPRE-luc) using Metafectene transfection reagent (Biontex, Martinsried/Planegg, Germany). The effect of GTS-21 on reporter gene activity was assessed by pre-treating the cells with GTS-21 before LPS stimulation (100 ng/mL; 6 h) after 36 h transfection. Following that, cell lysates were collected and a luciferase assay was performed as described previously [29].

### 4.12. Electrophoretic Mobility Shift Assay (EMSA)

BV2 cells were pretreated for 1 h with GTS-21 before being stimulated with LPS for 1 or 3 h. The cells’ nuclear extracts were prepared in the same manner as previously described [43]. T4 polynucleotide kinase (New England Biolabs, Beverly, MA, USA) was used to end-label double-stranded DNA oligonucleotides containing the NF-κB, ARE, and CRE consensus sequences. Nuclear proteins (10 μg) were incubated on ice for 30 min with a ^32^ P-labeled probe before being resolved on a 5% acrylamide gel and visualized by autoradiography.

### 4.13. Statistical Analysis

SPSS for Windows was used for statistical analysis (version 18.0; SPSS Inc., Chicago, IL, USA). A one-way analysis of variance was used to examine the differences between the groups (ANOVA). The least significant difference (LSD) test was used for multi-comparisons (post hoc). All values are given as the mean ± standard error of the mean (SEM). Statistical significance was indicated by a *p*-value less than 0.05.

## Figures and Tables

**Figure 1 ijms-23-04420-f001:**
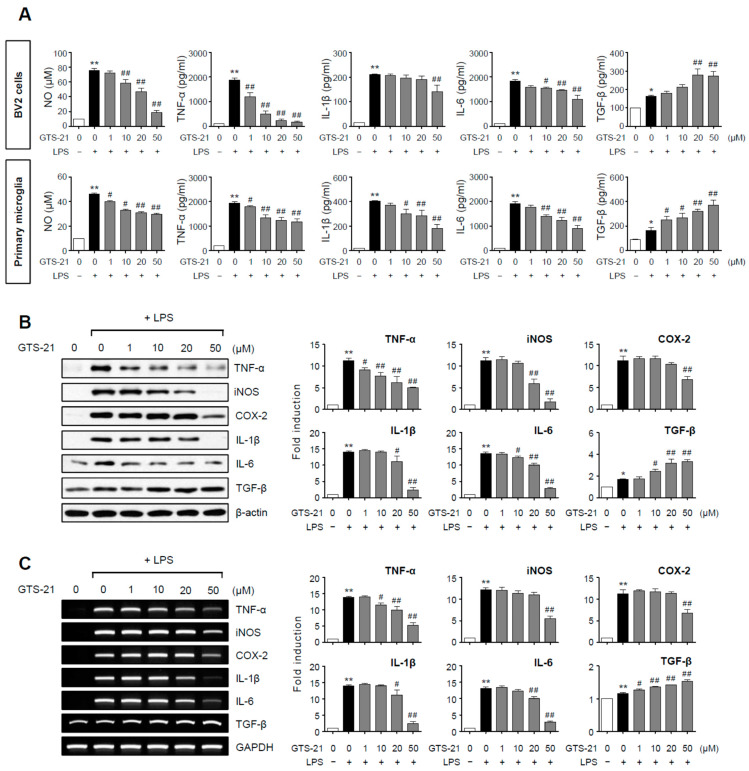
Effect of GTS-21 on inflammatory molecules in LPS-stimulated microglial cells. (**A**) After pretreatment with GTS-21 for 1 h, BV2 cells or primary cultured microglia were incubated with LPS (100 ng/mL for BV2, 10 ng/mL for primary microglia). The levels of nitrite, TNF-α, IL-1β, IL-6, and TGF-β were assessed in supernatants after a 16 h incubation period (*n* = 3–5 per group). (**B**,**C**) BV2 cells were treated with GTS-21 for 1 h before being incubated with LPS for 6 h. Western blot analysis (**B**) and RT-PCR (**C**) were performed to determine the expression level of inflammatory molecules (*n* = 3–4 per group). The left panel shows representative blots/gels, whereas the right panel shows quantitative data. The data are presented as the mean ± SEM. * *p* < 0.05, vs. control group; ** *p* < 0.01, vs. control group; ^#^
*p* < 0.05 vs. LPS-treated group; ^##^ *p* < 0.01 vs. LPS-treated group.

**Figure 2 ijms-23-04420-f002:**
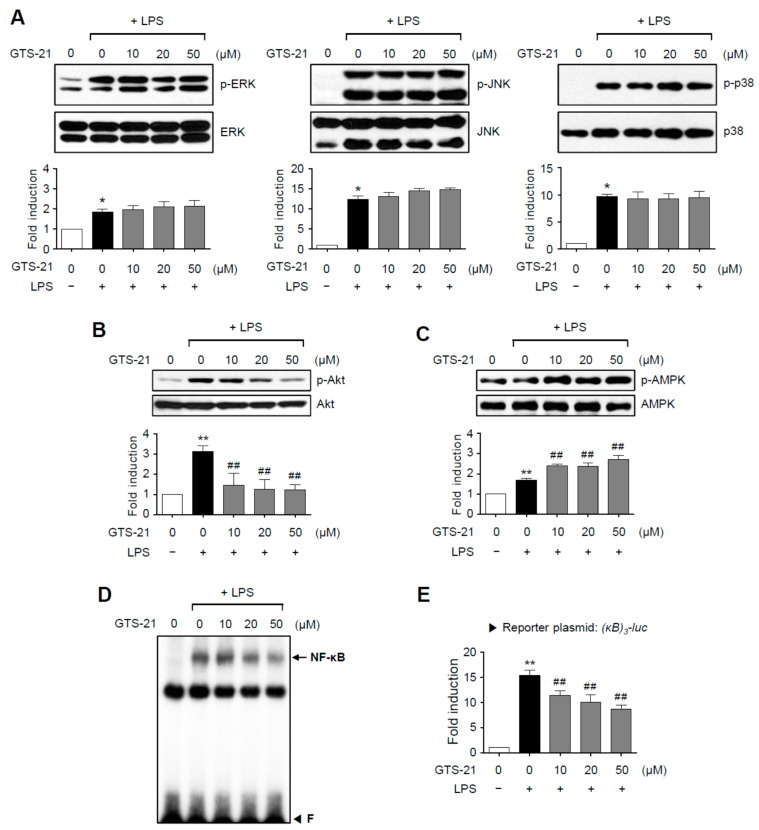
Effect of GTS-21 on the activities of MAPKs, Akt, AMPK, and NF-κB in LPS-stimulated BV2 cells. (**A**–**C**) Cell lysates from BV2 cells treated with LPS for 30 min in the presence or absence of GTS-21 were prepared, and western blot analysis was performed to investigate the effect of GTS-21 on MAPKs. (**A**), Akt (**B**), and AMPK (**C**) activity (*n* = 3–5 per group). The bottom panels show the quantification data. Phosphorylated MAPKs, Akt, and AMPK levels were adjusted to total forms and expressed as fold changes compared to untreated control samples. (**D**) Nuclear extracts from BV2 cells treated with GTS-21 in the presence of LPS for 1 h were used for EMSA for NF-κB. ‘F’ stands for free probe. (**E**) Analysis of the [κB]_3_-luc reporter gene activity after transient transfection (*n* = 5). The data are presented as the mean ± SEM. * *p* < 0.05, vs. control group; ** *p* < 0.01, vs. control group; ^##^ *p* < 0.01 vs. LPS-treated group.

**Figure 3 ijms-23-04420-f003:**
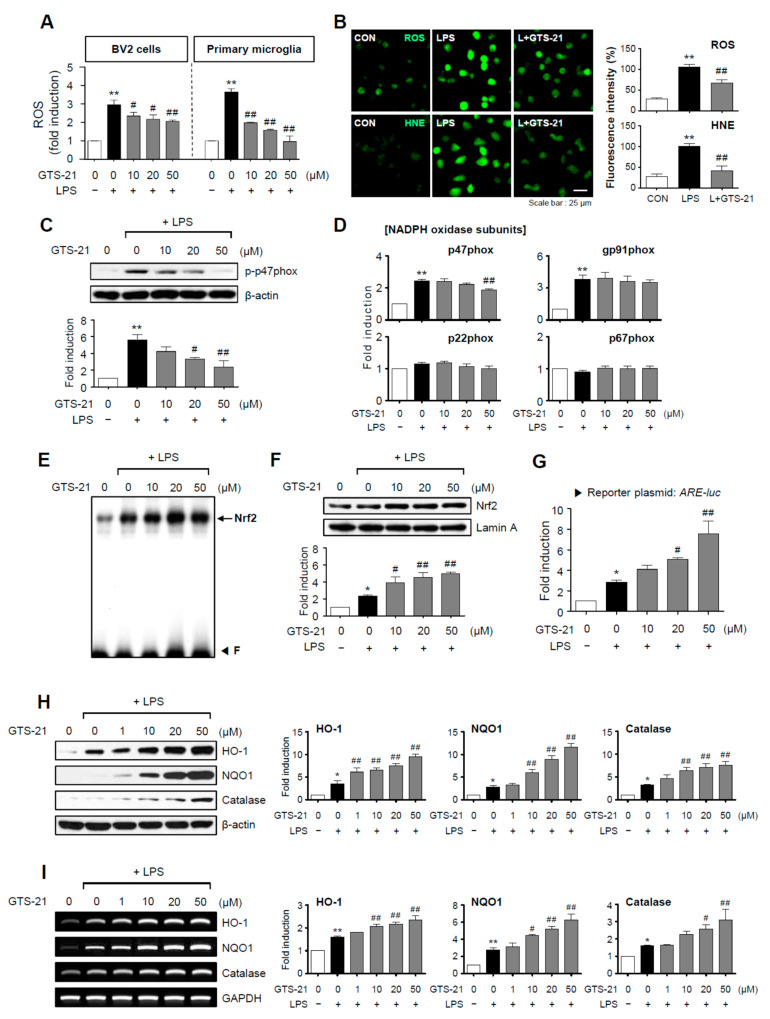
GTS-21 inhibited the production of ROS and HNE by inhibiting the NADPH oxidase subunit p47phox and increasing Nrf2/ARE signaling. (**A**) GTS-21 was applied to microglial cells 1 h before LPS stimulation for 16 h, and intracellular ROS levels were measured using the DCF-DA method (*n* = 4). (**B**) A representative confocal image of CellROX-derived fluorescence generated by intracellular ROS, and immunofluorescence staining for HNE in BV2 cells (*n* = 4). (**C**) Phosphorylation of the p47phox subunit was determined using western blot analysis (*n* = 3). (**D**) Quantitative RT-PCR to determiner mRNA expression level of NADPH oxidase subunits in BV2 microglia (*n* = 3–4 per group). (**E**) EMSA to assess the DNA binding activity of Nrf2. (**F**) The nuclear translocation of Nrf2 was assessed by western blot analysis (*n* = 3). (**G**) ARE-luc reporter gene activity after transient transfection (*n* = 3). (**H**,**I**) Effects of GTS-21 on the protein and mRNA expressions of *HO-1*, *NQO1*, and *catalase* in BV2 cells (*n* = 3). Western blot (**H**) and RT-PCR (**I**) data are shown. The left panel shows representative blots/gels, whereas the right panel shows quantitative data. The data are presented as the mean ± SEM. * *p* < 0.05, vs. control group; ** *p* < 0.01, vs. control group; ^#^ *p* < 0.05 vs. LPS-treated group; ^##^ *p* < 0.01 vs. LPS-treated group.

**Figure 4 ijms-23-04420-f004:**
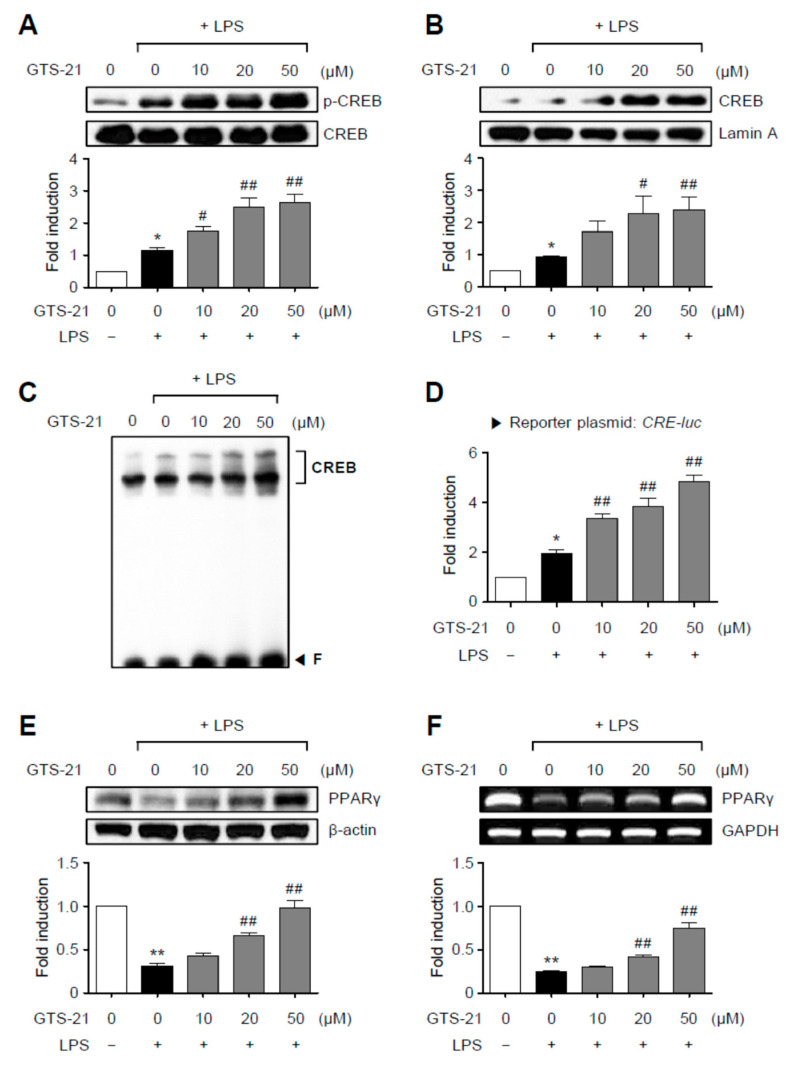
GTS-21 enhanced CREB and PPARγ signaling in LPS-stimulated BV2 cells. (**A**) BV2 cells were pretreated for 1 h with GTS-21 at the indicated concentrations before being stimulated with LPS for 30 min (*n* = 3). The levels of phospho- and total CREB were determined using western blot analysis on cell lysates. (**B**) CREB nuclear translocation was detected using a western blot (*n* = 4). Quantification data are shown at bottom. (**C**) Nuclear extracts from BV2 cells treated with GTS-21 in the presence of LPS for 1 h were used to perform EMSA for CREB. (**D**) CRE-luc reporter gene activity after transient transfection (*n* = 3). (**E**,**F**) BV2 cells were treated with GTS-21 for 1 h before being incubated with LPS for 6 h. To determine the effects of GTS-21 on the protein and mRNA expressions of *PPARγ*, western blot analysis (**E**) and RT-PCR (**F**) were performed (*n* = 3–4 per group). The upper panel shows representative blots/gels, while the bottom panel shows quantitative data. The data are presented as the mean ± SEM. * *p* < 0.05, vs. control group; ** *p* < 0.01, vs. control group; ^#^ *p* < 0.05 vs. LPS-treated group; ^##^ *p* < 0.01 vs. LPS-treated group.

**Figure 5 ijms-23-04420-f005:**
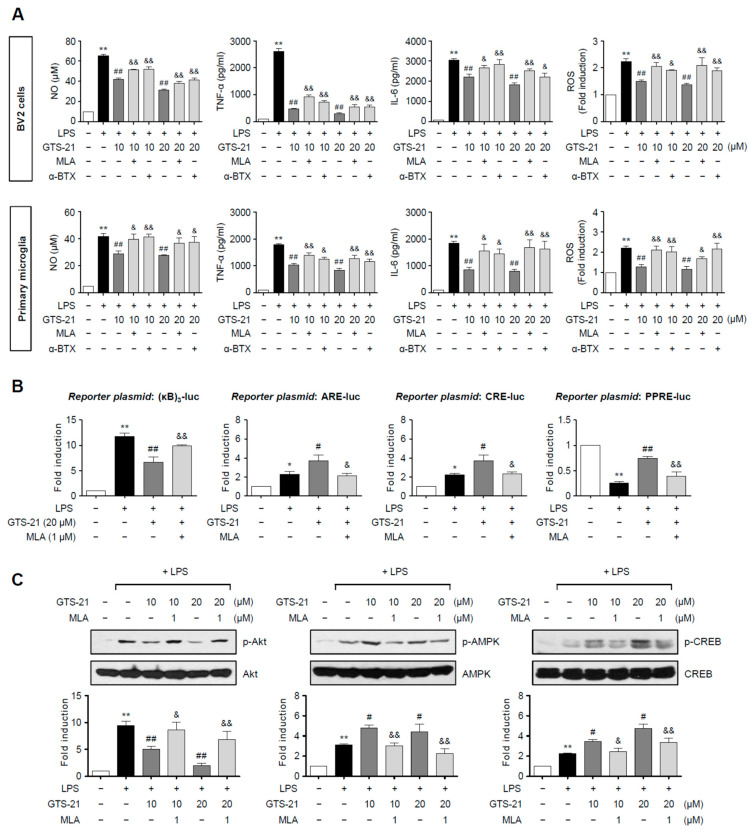
α7 nAChR antagonists reversed the anti-inflammatory effects of GTS-21 in LPS-stimulated microglia. (**A**) The effect of methyllycaconitine (MLA) and α-bungarotoxin (α-BTX) on the production of NO, TNF-α, IL-6, and ROS in LPS + GTS-21-treated BV2 cells and primary microglia (*n* = 3–4 per group). Cells were pre-treated with MLA (1 μM) or α-BTX (0.01 μM) for 1 h, and then GTS-21 (10, 20 μM) for 1 h, followed by the treatment with LPS (100 ng/mL for BV2, 10 ng/mL for primary microglia) for 16 h. The levels of NO, TNF-α, and IL-6 released into the medium, as well as intracellular ROS, were measured. (**B**) The effect of MLA on NF-κB, Nrf2, CREB, and PPARγ reporter gene activities (*n* = 3–4 per group). BV2 cells were transfected with the reporter plasmid and then treated with MLA (1 μM) for 1 h, followed by GTS-21 (20 μM) and LPS (100 ng/mL). Cells were harvested after 6 h of LPS treatment and the reporter gene assay was carried out. (**C**) BV2 cells were pre-treated with MLA (1 μM) for 1 h, and then GTS-21 (10, 20 μM) for 1 h, followed by the treatment with LPS for 30 min to determine p-Akt, p-AMPK, and p-CREB levels by western blot analyses (*n* = 3–4 per group). The data are presented as the mean ± SEM. * *p* < 0.05, vs. control group; ** *p* < 0.01, vs. control group; ^#^ *p* < 0.05 vs. LPS-treated group; ^##^ *p* < 0.01 vs. LPS-treated group; ^&^ *p* < 0.05 vs. LPS + GTS-21-treated group; ^&&^ *p* < 0.01 vs. LPS + GTS-21-treated group.

**Figure 6 ijms-23-04420-f006:**
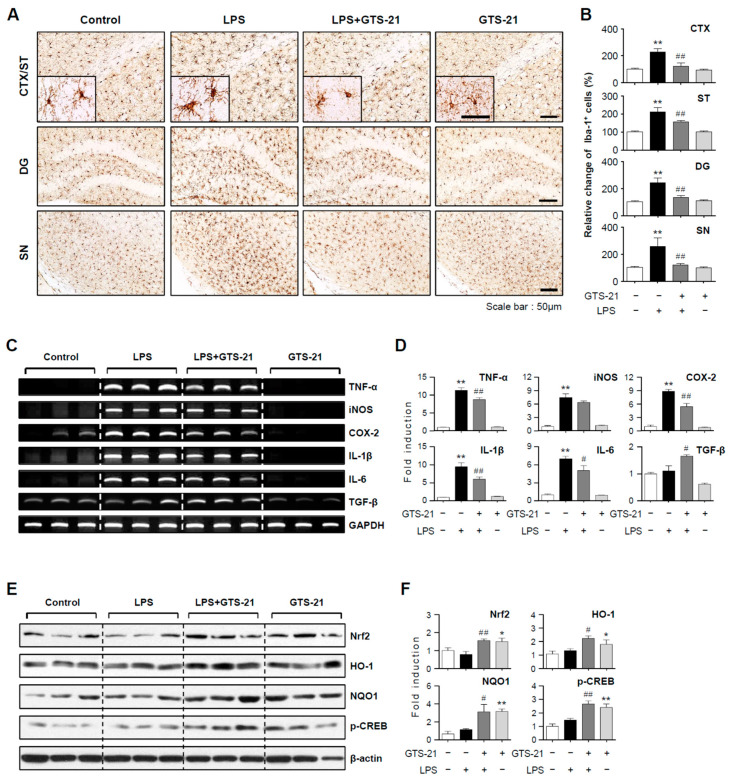
Effect of GTS-21 on microglial activation, inflammatory markers, and antioxidant signaling molecules in the brains of LPS-injected mice. (**A**,**B**) Iba-1 immunohistochemical staining and quantification of Iba-1-positive microglia 24 h after LPS injection (*n* = 4 per group, 3 sections/brain). GTS-21 (5 mg/kg) reduced microglial activation in the prefrontal cortex (CTX), striatum (ST), dentate gyrus of the hippocampus (DG), and substantia nigra (SN) of LPS-injected mice. Data quantification (**B**) and representative images (**A**) are shown. Scale bars, 50 μm. (**C**,**D**) Effects of GTS-21 on iNOS, COX-2, and cytokines mRNA levels in the cortices of LPS-injected mice (*n* = 3). The quantification data (**D**) and representative gels (**C**) are shown. (**E**,**F**) Western blot analysis was performed on protein extracts from the cortex of each group using Nrf2, p-CREB, HO-1, and NQO1 antibodies (*n* = 4–5 per group). Representative blots (**E**) and quantification data (**F**) are shown. The data are presented as the mean ± SEM. * *p* < 0.05, vs. control group; ** *p* < 0.01, vs. control group; ^#^ *p* < 0.05 vs. LPS-treated group; ^##^ *p* < 0.01 vs. LPS-treated group.

**Figure 7 ijms-23-04420-f007:**
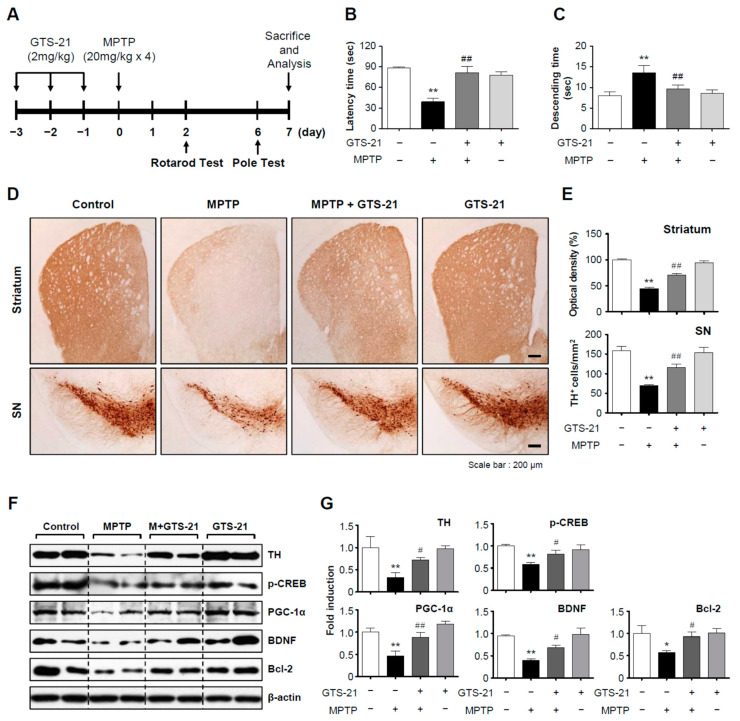
GTS-21 ameliorated the impaired movement and dopaminergic neuronal cell death in the brains of MPTP-injected mice. (**A**) The experimental procedure is depicted schematically. Mice were given GTS-21 (2 mg/kg, i.p.) daily for three days prior to MPTP injection. Mice were sacrificed 7 days after MPTP injection for histological and biochemical analyses. (**B**,**C**) The rotarod and pole tests were carried out two and six days after the MPTP injection, respectively (*n* = 8–9 per group). (**D**,**E**) TH-positive neuronal cells in the substantia nigra and striatum (representative pictures) (**D**). The optical density of TH-positive fibers in the striatum and the number of TH-positive cells in the substantia nigra were measured for quantitative analysis (**E**) (*n* = 4 per group, 3 sections/brain). (**F**,**G**) Protein extracts from the substantia nigra of each group were analyzed using TH, p-CREB, PGC-1, BDNF, and Bcl2 antibodies (*n* = 4). The figures show representative blots (**F**) and quantification data (**G**). The data are presented as the mean ± SEM. * *p* < 0.05, vs. control group; ** *p* < 0.01, vs. control group; ^#^ *p* < 0.05 vs. MPTP-treated group; ^##^ *p* < 0.01 vs. MPTP-treated group.

**Figure 8 ijms-23-04420-f008:**
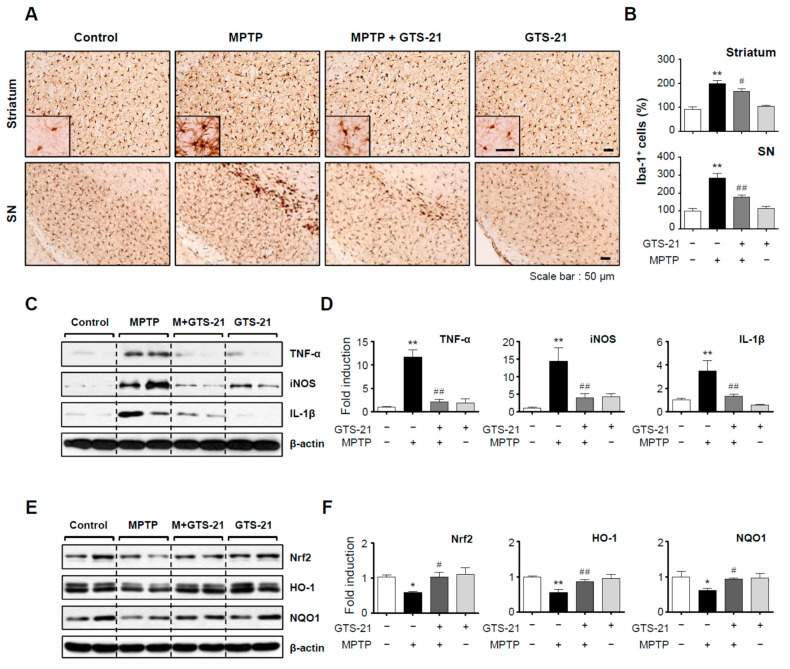
Effect of GTS-21 on microglial activation, inflammatory markers, and antioxidant enzyme expression in the brains of MPTP-injected mice. (**A**) Iba-1-positive microglial cells in the substantia nigra and striatum (representative images). (**B**) Quantitative analysis was performed by measuring the number of Iba-1-positive cells (*n* = 4 per group, 3 sections/brain). (**C**–**F**) Protein extracts from the substantia nigra of each group were analyzed using TNF-α, iNOS, and IL-1β antibodies or Nrf2, HO-1, and NQO1 antibodies (*n* = 4). The figures show representative blots (**C**,**E**) and quantification data (**D**,**F**). The data are presented as the mean ± SEM. * *p* < 0.05, vs. control group; ** *p* < 0.01, vs. control group; ^#^ *p* < 0.05 vs. MPTP-treated group; ^##^ *p* < 0.01 vs. MPTP-treated group.

**Table 1 ijms-23-04420-t001:** Primer sequences used for PCR.

Gene	Forward Primer (5′−3′)	Reverse Primer (5′−3′)	Size
*iNOS*	CAAGAGTTTGACCAGAGGACC	TGGAACCACTCGTACTTGGGA	450 bp
*TNF-α*	CCTATGTCTCAGCCTCTTCT	CCTGGTATGAGATAGCAAAT	354 bp
*IL-1β*	GGCAACTGTTCCTGAACTCAACTG	CCATTGAGGTGGAGAGCTTTCAGC	447 bp
*IL-6*	CCACTTCACAAGTCGGAGGCTT	CCAGCTTATCTGTTAGGAGA	395 bp
*COX-2*	TTCAAAAGAAGTGCTGGAAAAGGT	GATCATCTCTACCTGAGTGTCTTT	304 bp
*TGF-β*	GCAGGAGCGCACAATCATGT	GCCCTGGATACCAACTATTG	327 bp
*NQO1*	AGAGGCTCTGAAGAAGAGAGG	CACCCTGAAGAGAGTACATGG	401 bp
*HO-1*	ATACCCGCTACCTGGGTGAC	TGTCACCCTGTGCTTGACCT	209 bp
*Catalase*	CCTGACATGGTCTGGGACTT	CAAGTTTTTGATGCCCTGGT	245 bp
*PPARγ*	CCGAAGAACCATCCGATT	CGGGAAGGACTTTATGTA	271 bp
*p47phox*	CGATGGATTGTCCTTTGTGC	ATCACCGGCTATTTCCCATC	256 bp
*p67phox*	CCCTTGGTGGAAGTCCAAAT	ATCCTGGATTCCCATCTCCA	242 bp
*gp91phox*	ACTGCGGAGAGTTTGGAAGA	GGTGATGACCACCTTTTGCT	201 bp
*p22phox*	AAAGAGGAAAAAGGGGTCCA	TAGGCTCAATGGGAGTCCAC	239 bp
*GAPDH*	ATGTACGTAGCCATCCAGGC	AGGAAGGAAGGCTGGAAGAG	420 bp

## Data Availability

The datasets generated for this study are available on request to the corresponding authors.

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
