# Peer review of "Anti-Inflammatory and Neuroprotective Mechanisms of GTS-21, an α7 Nicotinic Acetylcholine Receptor Agonist, in Neuroinflammation and Parkinson’s Disease Mouse Models"

_ijms, 2022, doi:10.3390/ijms23084420_

Round 1

Reviewer 2 Report

In the present manuscript, Park et al., investigated the Anti-inflammatory and neuroprotective mechanisms of GTS- 21 in neuro-inflammation in Parkinson’s disease mouse models. The study revealed that α7 nAChR agonist GTS-21 exerts its anti-inflammatory and neuroprotective effects through the Nrf2/ARE and PKA/CREB pathways. The study was performed in LPS-induced systemic inflammation and MPTP-induced PD mice model pre-treated with GTS-21. The motor coordination of the experimental animals was evaluated using an accelerated rotarod test. Immortalized murine BV2 microglial cell line and primary microglial cells were used as an in vitro model to study the expression of iNOS, COX-2 in LPS-stimulated microglial cells. Western blot and RT PCR was performed for evaluating the expression level of inflammatory molecules. Iba-1 immunohistochemical staining was performed to understand the microglial activation. Dopaminergic neuronal cell death in the brain regions of MPTP-injected mice was analysed using Tyrosine hydroxylase staining.

  • 1-methyl-4-phenyl-1,2,3,6-tetrahydropyridine (MPTP) and 6-hydroxydopamine (6-OHDA) induced animals are the two widely used models for Parkinson’s disease. In the present study, the authors used MPTP induced Parkinson’s model. It's well-known that the MPTP model lacks aggregation of α-synuclein, which is a primary pathology of PD. On the other hand, long term 6-OHDA Parkinson’s model is observed with α-synuclein aggregation and selective loss of nigrostriatal dopaminergic neurons. Keeping in mind that the activation of α7-nAChRs promotes the clearance of α-Synuclein, what justification can authors provide for selecting the MPTP over 6-OHDA.
  • In the present study, the Authors did the GTS-2 Pre-treatment followed by LPS-induced systemic inflammation and MPTP-induced PD in mice. Interestingly, GTS-2 dosage and treatment duration for LPS-induced systemic inflammation and MPTP-induced PD mice were different. What’s the rationale behind this difference in the treatment dosage and duration?
  • Authors should check the font size in the figures. In figure 7 “Y-axis” font size is different

Author Response

1
Response to Reviewer 2 Comments
Point 1: 1-methyl-4-phenyl-1,2,3,6-tetrahydropyridine (MPTP) and 6-hydroxydopamine (6-OHDA)
induced animals are the two widely used models for Parkinson’s disease. In the present study, the
authors used MPTP induced Parkinson’s model. It's well-known that the MPTP model lacks
aggregation of α-synuclein, which is a primary pathology of PD. On the other hand, long term 6-
OHDA Parkinson’s model is observed with α-synuclein aggregation and selective loss of nigrostriatal
dopaminergic neurons. Keeping in mind that the activation of α7-nAChRs promotes the clearance of
α-Synuclein, what justification can authors provide for selecting the MPTP over 6-OHDA.
Response 1: The PD mouse model using MPTP has several merits: 1) convenience of applicability:
mice are simply intraperitoneally injected with MPTP, while 6-OHDA should be stereotaxically
injected under anesthesia; 2) relevant behavioral assessment: the MPTP-induced mouse model
manifested PD-like motor symptoms characterized by impaired motor coordination in the rotarod
test and askinesia in the pole test; and 3) apparent PD-like pathology: nigrostriatal degeneration and
sustained neuroinflammation were observed in the acute MPTP mouse model, although α-synuclein
aggregation was not observed. Because our study focused on the anti-inflammatory role of GTS-21
and its mechanism of action under PD-like conditions, the acute MPTP model was thought to be
appropriate for investigating the effects of GTS-21 against MPTP neurotoxicity. Recently, our group
also established a subacute MPTP mouse model of PD, in which α-synuclein aggregation was
observed (Biomed Pharmacother. 2020, 130:110576). Thus, it would be interesting to observe the effect
of GTS-21 on α-synuclein aggregation in this model in the future.
Point 2: In the present study, the Authors did the GTS-2 Pre-treatment followed by LPS-induced
systemic inflammation and MPTP-induced PD in mice. Interestingly, GTS-2 dosage and treatment
duration for LPS-induced systemic inflammation and MPTP-induced PD mice were different. What’s
the rationale behind this difference in the treatment dosage and duration?
Response 2: In a preliminary study, GTS-21 was administered at a dose of 2 or 5 mg/kg to MPTPinduced PD mice. The results from 2 mg/kg treatment of GTS-21 were more consistent than those
from 5 mg/kg in MPTP-treated mice. Further, the results obtained with 5 mg/kg fluctuated
significantly. Therefore, mice were treated with 2 mg/kg GTS-21 in the MPTP-induced PD model.
Regarding the dosage of GTS-21 in LPS-induced systemic inflammation mice, 5 or 15 mg/kg of GTS21 was administered in our preliminary experiment. Several pro-inflammatory responses (microglial
population and activation and pro-inflammatory cytokine levels) in 5 mg/kg GTS-21-administered
mice were comparable with those of 15 mg/kg GTS-21-administered mice. These effects of GTS-21
were thought to be saturated at 5 mg/kg dose in LPS-treated brains.; therefore, we administered 5
mg/kg GTS-21 in LPS-treated mice. The difference in the GTS-21 dosage between MPTP- and LPStreated mice may be partly due to higher MPTP toxicity than LPS .
In this context, GTS-21 treatment was applied differently to MPTP and LPS models. Mice were pretreated with GTS-21 for 3 days before MPTP injection without additional treatment with the
2
compound immediately before MPTP injection because additional treatment may produce a high rate
of mortality in the acute MPTP model. Although the exact cause is unknown, it may be due to the
sudden amplification of MPTP-induced toxic shock. In contrast, pretreatment with GTS-21 for 4 days,
including an additional injection of the compound immediately before LPS exposure, did not induce
toxicity in a single bout of the LPS model. Thus, the duration of GTS-21 pretreatment was conveyed
differently in the LPS and MPTP models in our experimental paradigm.
Point 3: Authors should check the font size in the figures. In figure 7 “Y-axis” font size is different.
Response 3: We have corrected the font size in Figure 7 of the revised manuscript.